# Genomic Bootstrap Barcodes and Their Application to Study the Evolution of Sarbecoviruses

**DOI:** 10.3390/v14020440

**Published:** 2022-02-21

**Authors:** Alexandre Hassanin, Opale Rambaud, Dylan Klein

**Affiliations:** Institut de Systématique, Évolution, Biodiversité (ISYEB), Sorbonne Université, Centre National de la Recherche Scientifique, École Pratique des Hautes Études, Muséum National d’Histoire Naturelle, Université des Antilles, 75231 Paris, France; opale.rambaud@gmail.com (O.R.); klein.dylan@outlook.com (D.K.)

**Keywords:** RNA recombination, COVID-19 origin, reservoir host, viral circRNA, phylogenetic support, tree reconstruction

## Abstract

Recombination creates mosaic genomes containing regions with mixed ancestry, and the accumulation of such events over time can complicate greatly many aspects of evolutionary inference. Here, we developed a sliding window bootstrap (SWB) method to generate genomic bootstrap (GB) barcodes to highlight the regions supporting phylogenetic relationships. The method was applied to an alignment of 56 sarbecoviruses, including SARS-CoV and SARS-CoV-2, responsible for the SARS epidemic and COVID-19 pandemic, respectively. The SWB analyses were also used to construct a consensus tree showing the most reliable relationships and better interpret hidden phylogenetic signals. Our results revealed that most relationships were supported by just a few genomic regions and confirmed that three divergent lineages could be found in bats from Yunnan: *SCoVrC*, which groups SARS-CoV related coronaviruses from China; *SCoV2rC*, which includes SARS-CoV-2 related coronaviruses from Southeast Asia and Yunnan; and *YunSar*, which contains a few highly divergent viruses recently described in Yunnan. The GB barcodes showed evidence for ancient recombination between *SCoV2rC* and *YunSar* genomes, as well as more recent recombination events between *SCoVrC* and *SCoV2rC* genomes. The recombination and phylogeographic patterns suggest a strong host-dependent selection of the viral RNA-dependent RNA polymerase. In addition, SARS-CoV-2 appears as a mosaic genome composed of regions sharing recent ancestry with three bat *SCoV2rC*s from Yunnan (RmYN02, RpYN06, and RaTG13) or related to more ancient ancestors in bats from Yunnan and Southeast Asia. Finally, our results suggest that viral circular RNAs may be key molecules for the mechanism of recombination.

## 1. Introduction

Many RNA viruses are known to evolve through recombination, a process resulting in mosaic genomes containing regions from different parental viruses. A copy-choice model (also named template switching model) has been proposed to account for recombination in unsegmented RNA viruses: during RNA replication, the viral RNA-dependent RNA polymerase (RdRp) can pause on the RNA template and switch to another template during synthesis, thereby generating a recombinant RNA molecule with mixed ancestry [1]. RNA recombination is assumed to be rare because it can only happen when a host cell is co-infected by at least two genetically distinct viruses. However, this process can generate a high diversity of recombinant viruses, and natural selection can favor some of them to adapt to new environments and hosts [1,2].

RNA recombination seems particularly active in RNA viruses of the subgenus *Sarbecovirus* [3,4], the taxonomic group including SARS-CoV and SARS-CoV-2, which are the causal agents involved in the severe acute respiratory syndrome (SARS) epidemic and coronavirus disease 2019 (COVID-19) pandemic, respectively. Although two sarbecoviruses were recently discovered in Sunda pangolins (*Manis javanaica*), all major lineages of *Sarbecovirus* have been detected in horseshoe bats (Chiroptera, Rhinolophidae, *Rhinolophus*) [5,6]. Most bat sarbecoviruses currently described were identified from various *Rhinolophus* species captured in caves of several provinces of China [7]. In addition, a few sarbecoviruses were detected in *Rhinolophus* species from Southeast Asia [8,9], Japan [10], Europe [11], and Africa [12], suggesting that horseshoe bats of the Old World constitute the natural reservoir host in which sarbecoviruses have evolved and diversified for several centuries [3,5,6]. Interestingly, Southeast Asia is a hot spot for the biodiversity of *Rhinolophus* as more than half of the species occur in this region [13]. Despite the limited number of field expeditions conducted in Southeast Asia, two SARS-CoV-2 related coronaviruses (*SCoV2rC*s) have already been described: one from two *R. shameli* of Cambodia and the other from five *R acuminatus* of Thailand [8,9]. In China, bat sampling efforts have been very important for the past fifteen years, and SARS-CoV related coronaviruses (*SCoVrC*s) were collected mainly in *R. sinicus* and also, less frequently, in several other bat species. However, a higher diversity of sarbecoviruses has been described from bats of the Yunnan province, with three divergent lineages: (i) SCoVrCs were collected in *R. sinicus* and several other species, such as *R. affinis*, *R. malayanus*, *R. nippon, R. pusillus*, and *R. stheno* [7]; (ii) SCoV2rCs were detected in the three species *R. affinis*, *R. malayanus*, and *R. pusillus* [14,15,16] and (iii) several divergent viruses, included in the *YunSar* group by Hassanin [17], were recently described in the four species *R. affinis*, *R. malayanus*, *R. pusillus*, and *R. stheno* [16,18] (but see paragraph 4.2). As shown in Hassanin [17], these three divergent lineages exhibit different synonymous nucleotide compositions, suggesting that most of their evolution took place in separate *Rhinolopus* species assemblages, with *YunSar* in Yunnan, *SCoVrC* in China, and *SCoV2rC* in Southeast Asia (Cambodia, Laos, Thailand, and Vietnam) [6]. However, genomic recombination between divergent viruses may have occurred in the caves of Yunnan, where the three lineages can be found in sympatry.

Genomic recombination can be highly misleading for inferring phylogenetic relationships. For instance, conflicting trees have been recently published using full-length genome alignments of sarbecoviruses: in Zhou et al. [16], three *YunSar* genomes (RmYN05, RmYN08, and RstYN04) were found to group with the pangolin virus from Guangxi; whereas, in Guo et al. [18], eight *YunSar* genomes (RaTG15, Rst7896, Rst7905, Rst7907, Rst7921, Rst7924, Rst7931, and Rst7952) appeared as the sister-group of the clade uniting *SCoVrC*, *SCoV2rC*, and the bat virus from Japan (Rc-o319, [10]). Such topological discordance was not expected as the three *YunSar* genomes of Zhou et al. [16] share 98% of genome identity with the eight *YunSar* genomes of Guo et al. [18]. Although the two trees were reconstructed using the maximum likelihood (ML) method, Zhou et al. [16] used the RAxML software and a GTR model, whereas Guo et al. [18] used the MEGA6 software and a Jukes-Cantor model. Therefore, the conflicting trees published by Zhou et al. [16] and Guo et al. [18] could be explained by the use of different models of nucleotide evolution, which may manage differently recombinant viral genomes during tree reconstruction.

To better interpret conflicting phylogenetic signals due to recombination, we report hereinafter a sliding window bootstrap (SWB) method for generating genomic bootstrap barcodes (GB barcodes) to highlight the regions of a genome alignment supporting phylogenetic relationships. The method was applied to a multiple alignment of 56 sarbecoviruses sampled from 52 bats, two pangolins, and two humans (SARS-CoV and SARS-CoV-2). The SWB analyses were also used to evidence the most reliable phylogenetic relationships using the SuperTRI method [19]. Our five main aims were: (i) to design a new tool to visualize the genomic regions supporting phylogenetic relationships; (ii) to reveal and interpret hidden phylogenetic signals; (iii) to provide a more reliable method of tree reconstruction for studying the evolutionary history of sarbecoviruses; (iv) to better characterize diversifying recombination between divergent *Sarbecovirus* lineages; and (v) to improve our knowledge on the mechanism of genomic recombination between RNA viruses.

## 2. Materials and Methods

### 2.1. DNA Alignment of Genomic Sequences

Full genomes of *Sarbecovirus* available in June 2021 in GenBank (https://www.ncbi.nlm.nih.gov/ accessed on 1 June 2021), GISAID (https://www.epicov.org/ accessed on 1 June 2021), and NGDC (https://ngdc.cncb.ac.cn/ accessed on 1 June 2021) databases were downloaded in Fasta format. Sequences with a large stretch of missing data were removed. Only a single sequence was retained for similar genomes showing less than 0.1% of nucleotide divergence, such as those available for human SARS-CoV-2 (millions of sequences), pangolin viruses from Guangxi (5 sequences), bat viruses from Cambodia (two sequences), etc. The details on the 56 selected genomes are provided in Appendix A. They include all viral lineages previously described within the subgenus *Sarbecovirus*. The 56 genomes were aligned in Geneious Prime^®^ 2020.0.3 (Biomatters Ltd., Auckland, New Zealand) with MAFFT version 7.450 [20] using default parameters. Then, the alignment was corrected manually on AliView 1.26 [21] based on translated and untranslated nucleotide sequences using the three following criteria: (i) the number of indels was minimized because they are rarer events than amino-acid or nucleotide substitutions; (ii) changes between similar amino-acids were preferred (using the ClustalX color scheme available in AliView); and (iii) transitions were privileged over transversions because they are more frequent. The insertion(s) found in only one virus and the regions with large gaps (>50 nucleotides (nt)) were removed from the multiple genome alignment to guarantee enough phylogenetic signal for the SWB analysis based on the smallest window size (i.e., 250 nt; see next paragraph). For that reason, the open reading frame 8 (ORF8) was excluded from the alignment because it is missing in the two sequences here used as outgroup, i.e., RspKY72 and RbBM48-31 [11,12].

### 2.2. Construction of Genomic Bootstrap Barcodes

The python scripts of SWB and BBC programs were specially written to construct GB barcodes. The SWB program was designed to conduct bootstrap analyses [22] on N subdatasets extracted from a multiple genome alignment (input file in Fasta format) using a window of W nucleotides (W parameter) moving in steps of S nucleotides (S parameter) along the alignment. Using W = 250 nt and S = 50 nt, the alignment of 28,845 nt was transformed into N = 572 overlapping subdatasets (or windows). In the SWB program, each of the 572 subdatasets of 250 nt were executed in RAxML [23] with a GTR+G model and 100 bootstrap replicates using the ML method. The phylogenetic results were then summarized in the SWB output, a CSV file containing the bootstrap percentages (BP) calculated from each of the 572 subdatasets for the 244,316 bipartitions (nodes) reconstructed during the SWB_250_ analysis. Three other SWB analyses were performed using the same S parameter (i.e., 50 nt) but different window sizes, i.e., W = 500 nt (SWB_500_, N = 567), 1000 nt (SWB_1000_, N = 557), or 2000 nt (SWB_2000_, N = 537).

The BBC program was written to construct the GB barcodes of a selection of bipartitions (Figure 1). In the first step, only bipartitions supported by BP ≥ 70% in one or more bootstrap analyses of the N subdatasets were selected. In this way, we further examined only bipartitions showing a robust phylogenetic signal in at least one region (window) of the alignment. By contrast, the bipartitions with no robust phylogenetic signal, i.e., with BP < 70% in all bootstrap analyses of the subdatasets, were removed. Then, GB barcodes were constructed only for the 294 bipartitions shared between the four SWB analyses based on different window sizes.

For each selected bipartition, four GB barcodes were constructed with the four SWB files based on different window sizes. A GB barcode can be viewed as a simplified representation of the SWB results. With W = 250 nt, the alignment of 28,845 nt was separated into N = 572 overlapping windows of 250 nt, which were used for bootstrap analysis. In the genome alignment, the first window bootstrap analysis represents its 5′ terminal region, whereas the 572nd window bootstrap analysis represents its 3′ terminal region. A given bipartition can be supported by different BP values (from 0 to 100) in the 572 window bootstrap analyses. After calculating the median positions (pos.) of all the 572 windows, the 294 GB_250_ barcodes were constructed by replacing the intervals between two positions (5′ and 3′ windows) by one of the following colored bars: red if BP_5′_ ≤ 30%, grey if 30% < BP_5′_ < 70%, and green if BP_5′_ ≥ 70%. We proceeded similarly for GB_500_, GB_1000_, and GB_2000_ barcodes.

The alignment positions of the genomic regions containing robust phylogenetic signal (GRPS) were deduced from the SWB_2000_ output (SWB analysis based on a sliding window of 2000 nt). For each bipartition, the 5′- and 3′-ends of GRPS were defined by the positions of the 5′- and 3′-terminal windows showing BP_2000_ ≥ 70%. To be conservative, the extension of GRPS was conducted by accepting BP_2000_ values between 50 and 70%. Pairwise nucleotide distances were calculated for all GRPS detected in GB barcodes shown in Section 3. The software PAUP version 4.0a (Sinauer Associates: Sunderland, MA, USA) [24] was used to calculate p distances, and the data were exported in Microsoft^®^ Excel (Microsoft, Albuquerque, NM, USA) to analyze maximum and minimum values. By comparing the alignment positions of GRPSs between nested bipartitions, we constructed a network showing interactions between ’nested GB barcodes’. Considering two bipartitions linked by nested phylogenetic relationships, with the descendant bipartition B + C phylogenetically nested within the parental bipartition A + B + C, the two GB barcodes constructed for these two bipartitions were identified as ’nested GB barcodes’ only if they were found to share overlapping GRPS positions.

### 2.3. SuperTRI Analyses

The bootstrap bipartitions generated from the four SWB analyses based on different window sizes (250, 500, 1000, and 2000 nt) were used for SuperTRI analyses [19] to construct the trees showing the most reliable phylogenetic relationships. The LFG program was written to convert the SWB output file into bootstrap log files, which were then used as inputs in SuperTRI v57 [19] to construct an MRP (Matrix Representation with Parsimony) file. For example, the SWB_250_ file generated using a window of 250 nt was converted with the LFG program (python script) into N = 572 bootstrap log files (lists of bootstrap bipartitions), and these files were further transformed into an MRP file using SuperTRI v57. In the MRP_250_ file, each of the 614,634 characters represents a bipartition with its assigned BP calculated in one of the 572 window bootstrap analyses. The MRP_250_ file was then executed in PAUP version 4.0a [24] using 1000 bootstrap replicates of weighted parsimony (with bootstrap percentages assigned as weights) to construct the SuperTRI bootstrap 50% majority-rule consensus (SB_250_) tree (see [19] for more details on the method). Finally, the mean bootstrap percentage (MBP) and reproducibility index (proportion of the N bootstrap analyses supporting the bipartition) were calculated automatically in SuperTRI v57 for all nodes of the SB trees. The MBP of hidden phylogenetic relationships (i.e., bipartitions not shown in the SB trees) were calculated using the SWB output files for all GB barcodes shown in Section 3.

## 3. Results

### 3.1. Phylogenetic Signal in Windows Moving along the Alignment of Sarbecovirus Genomes

In this study, 56 genomes of the subgenus *Sarbecovirus* (Appendix A) were aligned to infer phylogenetic relationships. The positions of the coding sequences were the following in our final alignment of 28,845 nucleotides (nt): 190–21,196 for ORF1ab, including the RNA-dependent RNA polymerase gene (*RdRp*) at positions 13,096–15,885; 21,198–24,836 for the spike (*S*) gene; 24,845–25,669 for ORF3a; 25,694–25,924 for the envelope (*E*) gene; 25,975–26,643 for the membrane (*M*) gene; 26,654–26,845 for ORF6; 26,846–27,214 for ORF7a; 27,211–27,318 for ORF7b; 27,335–28,603 for the nucleocapsid (*N*) gene; and 28,628–28,744 for ORF10.

The dataset was bootstrapped using a window of W nucleotides (W parameter) moving in steps of 50 nt along the genome alignment in order to examine the distribution of the phylogenetic support (SWB program, Figure 1). The window size is a key parameter for bootstrap analyses because the amount of phylogenetic signal depends on both the number of nucleotide sites and their evolutionary rates. For that reason, we decided to perform four SWB analyses with the same step parameter of 50 nt but using four different window sizes, i.e., 250 nt, 500 nt, 1000 nt, or 2000 nt. The smallest window size (W = 250 nt) was applied to detect possible changes in phylogenetic relationships due to the recombinant origin of small genomic regions, whereas the largest window size (W = 2000 nt) was used to guarantee enough phylogenetic signal (i.e., informative sites) for bootstrap analyses. The two intermediate window sizes (500 and 1000 nt) were used to better interpret possible differences between the results based on the two extreme values.

The number of phylogenetically informative sites (IS) was firstly compared between the four SWB analyses (Figure 2A): the 537 windows of 2000 nt contain between 557 and 1372 IS, which is much more than the 557 windows of 1000 nt (between 264 and 769 IS), the 567 windows of 500 nt (between 123 and 404 IS), and the 572 windows of 250 nt (between 47 and 217 IS). In addition, the amount of phylogenetic signal appeared to be variable along the alignment, and this was particularly evident for the analysis based on the largest window size (2000 nt), for which the greatest number of IS (1372) was found in window N° 425 (pos. 21,201–23,200, *S* gene), while the smallest number of IS (557) was found in windows N° 282 and N° 283 (pos. 14,051–16,100, including the *RdRp* gene). These results, therefore, confirmed that the *S* gene evolves faster than other protein-coding genes of the genome [25].

The number of IS is expected to impact the number of bootstrap bipartitions. Indeed, a weak phylogenetic signal (i.e., low number of IS) can generate many possible phylogenetic hypotheses, i.e., several bipartitions supported by very low BP, whereas a strong phylogenetic signal (i.e., a high number of IS) can provide support for one specific phylogenetic hypothesis, i.e., one bipartition supported by BP ≥ 70%. This was confirmed by the results of Figure 2B, as the SWB_2000_ analysis produced fewer bipartitions (between 118 and 712) than the SWB_250_ analysis (between 360 and 2350). As a consequence, we conclude that the windows of 2000 nt contain enough phylogenetic signal for studying relationships at different depths (both shallow and deep evolutionary relationships), whereas the windows of 250 nt can be less efficient for studying shallow relationships (i.e., the most recent divergences) due to insufficient amount of phylogenetic signal (not enough available IS).

### 3.2. Topological Congruence between the Four SuperTRI Bootstrap Consensus Trees

The four SuperTRI bootstrap consensus (SB) trees reconstructed from the SWB analyses based on different window sizes (250, 500, 1000, and 2000 nt) are available in Appendix A. Most nodes of the trees were supported by maximum SuperTRI bootstrap values (SBP = 100). Since a strong overall topological congruence was found between the four SB trees, we decided to show in Figure 3 a 75%-majority consensus tree, in which topological differences between SB_250_, SB_500_, SB_1000_, and SB_2000_ trees are indicated by dash branches. As previously found by Ropiquet et al. [19], our SuperTRI analyses confirmed that the reproducibility index (*Rep*) (here the proportion of the N bootstrap analyses supporting a bipartition) is linearly correlated with the mean branch support (here MBP, calculated using all N bootstrap analyses) (Appendix A). Since it was much easier to calculate MBP than *Rep* values for hidden phylogenetic relationships, only MBP values were used in this study.

In the synthesis tree of SuperTRI analyses (Figure 3), three main clades of *Sarbecovirus* were found monophyletic with high MBP values, indicating that they were supported by most overlapping subdatasets (windows) extracted from the multiple genome alignment: (i) the western clade (62% ≤ MBP ≤ 98%), grouping RspKY72 from Africa and RbBM48-31 from Europe; (ii) the *SCoVrC* clade (28% ≤ MBP ≤ 47%), which includes all *SCoVrC*s found in different provinces of China; and (iii) a composite clade (28% ≤ MBP ≤ 46%), which contains the Japanese virus Rc-o319, the four *YunSar* viruses (RaTG15, RmYN05, Rst7931, and RstYN04), all *SCoV2rCs*, the two pangolin viruses (MjGuangdong and MjGuangxi), and the three *RecSar* viruses (showing evidence of past recombination between *SCoVrC* and *SCoV2rC*, i.e., RpPrC31, RsZXC21, and RsZC45; see paragraph 4.2). The GB barcodes associated with these three main *Sarbecovirus* clades (Figure 3) showed that the composite and *SCoVrC* clades were mainly supported by the 5′ region of the genome alignment, whereas the western clade was highly supported by all regions.

Within the composite clade, *YunSar* was found monophyletic with very high MBP values (≥98%), indicating that the phylogenetic signal was distributed in all regions of the alignment (see GB barcode N° 2 in Figure 3). Other relationships were less supported (MBP < 50%), including *SCoV2rC* (14% ≤ MBP ≤ 31%), *RecSar* (14% ≤ MBP ≤ 22%), the clade uniting MjGuangdong, *SCoV2rC* and *RecSar* (23% ≤ MBP ≤ 45%), and the clade herein referred to as *SCoV2rC sensu lato* (*s.l.*), which is composed of *SCoV2rC*, *RecSar* and the two pangolin viruses (19% ≤ MBP ≤ 36%).

Within the *SCoVrC* clade, we found a basal dichotomy separating two geographic groups: (i) *SCoVrC*-SW, which includes SARS-CoV and bat viruses collected in the provinces of Southwest China (Guangxi, Guizhou, Sichuan, and Yunnan) (10% ≤ MBP ≤ 20%); and (ii) *SCoVrC*-CE, which includes bat viruses collected in the provinces of Central and East China (Henan, Hong-Kong, Hubei, Shaanxi, and Zhejiang) (4% ≤ MBP ≤ 8%).

Since *RecSar* and *YunSar* have originated from recombination between divergent *Sarbecovirus* lineages (see paragraph 4.2 for details), the four SWB analyses were also carried out by removing these two groups (reduced alignment of 49 genomes). The synthesis tree of SuperTRI analyses (Appendix A) was found to be very similar to that of Figure 3. In particular, all nodes were recovered monophyletic with higher MBP values, including the western clade (66% ≤ MBP ≤ 99%), *SCoVrC* (38% ≤ MBP ≤ 74%), *SCoVrC*-CE (9% ≤ MBP ≤ 32%), *SCoVrC*-SW (11% ≤ MBP ≤ 40%), the composite clade (46% ≤ MBP ≤ 83%), *SCoV2rC s.l.* (48% ≤ MBP ≤ 76%), *SCoV2rC* + MjGuangdong (48% ≤ MBP ≤ 75%), *SCoV2rC* (30% ≤ MBP ≤ 62%), the monophyly of *SCoV2rCs* from Yunnan (*SCoV2rC*-YU; 17% ≤ MBP ≤ 40%), and the sister-group relationship between SARS-CoV-2 and RmYN02 + RpYN06 (15% ≤ MBP ≤ 31%).

### 3.3. Comparison between GB Barcodes Constructed Using Four Sliding Window Sizes

As shown in Figure 1, 294 bipartitions were recovered in all the four SWB analyses based on different window sizes. Some of these bipartitions correspond to the tree nodes of Figure 3, but most of them represent hidden phylogenetic relationships, i.e., less reliable phylogenetic hypotheses (not shown in the tree) generally supported by smaller genomic regions. For convenience, they were numbered from 1 to 294 after a classification based on decreasing values of MBP_2000_ (from 100 to 1%; calculated using the N = 572 bootstrap analyses performed with a window of 2000 nt). The four SWB output files reduced to the 294 bipartitions were then used as inputs in the BBC program to construct 294 × 4 = 1176 GB barcodes.

A barcode can be defined as a small image in which numbers, letters and/or other symbols were coded to represent a series of vertical bars of varying width. In our study, the GB barcodes were constructed for the 294 selected bipartitions to summarize the results of SWB analyses along the genome alignment. A GB barcode is a small image representing the genome alignment and in which the N BP values (N = 572 with a window size of 2000 nt) obtained for a bipartition (i.e., a phylogenetic hypothesis) were transformed into N colored vertical bars using the following code: green for BP ≥ 70%; grey for 70% > BP > 30%; and red for BP ≤ 30%. In a GB barcode, the genomic region(s) containing robust phylogenetic signal (hereinafter referred to as GRPS) for the bipartition of interest are therefore shown in green, whereas the region(s) of the genome alignment with no robust support are shown in red.

The GB_2000_ barcodes were constructed from the SWB_2000_ analysis, and they were compared to GB_1000_, GB_500,_ and GB_250_ barcodes built from the SWB_1000_, SWB_500_, and SWB_250_ analyses, respectively. As expected, the MBP and maximum window BP values were found to be higher for the SWB_2000_ analysis, i.e., based on the largest window size (2000 nt): an increase was found for 77%, 84%, and 87% of the 294 bipartitions when the MBP_2000_ values were compared to those calculated from the SWB analyses based on window sizes of 1000 nt (MBP_1000_), 500 nt (MBP_500_) and 250 nt (MBP_250_), respectively. A similar trend was found for maximum window BP values: an increase was found for 72, 74, and 84% of the 294 bipartitions when the maximum window BP_2000_ values were compared to those calculated from the SWB_1000_, SWB_500,_ and SWB_250_ analyses, respectively.

Some genomic regions supported by BP ≥ 70% (green regions) in GB_2000_ barcodes can be absent or much more reduced in GB_1000_, GB_500_, and GB_250_ barcodes (e.g., bipartitions N° 6, 8, and 18 in Figure 3). These differences can be explained by the limited amount of phylogenetic signal (number of IS) in some windows of 1000, 500, 250 nt. However, using the largest window size is not necessarily the panacea, as an incongruent phylogenetic signal located in a genomic region much smaller than the window size cannot be detected. This is the case for several bipartitions for which small regions supported by BP ≥ 70% in GB_1000_, GB_500_, and/or GB_250_ barcodes (in green) were found hidden (i.e., in red, BP ≤ 30%) in GB_2000_ barcodes (e.g., bipartitions N° 11, 22, and 63 in Figure 3). This explanation may also hold for several other bipartitions for which small regions showing no robust signal (BP ≤ 30%) in GB_1000_, GB_500_, and/or GB_250_ barcodes were found hidden (i.e., in green this time) in GB_2000_ barcodes (e.g., bipartitions N° 3, 4 and 9 in Figure 3). Using the smallest window size (250 nt) can be therefore crucial to detect changes in phylogenetic relationships due to the recombinant origin of small genomic fragments.

Four categories of GB barcodes were defined according to the genomic location of the phylogenetic support: (i) GB barcodes S5, in which the phylogenetic support was located in the 5′ region, i.e., the first third part of the genome alignment; (ii) GB barcodes SC, in which the phylogenetic support was located in the central region, i.e., the second third part of the genome alignment; (iii) GB barcodes S3, in which the phylogenetic support was located in the 3′ region, i.e., the last third part of the genome alignment; and (iv) other GB barcodes in which the phylogenetic support was distributed in several regions of the alignment. Several examples are described in the next three paragraphs.

### 3.4. GB Barcodes of Bipartitions Including YunSar Viruses

All the 15 GB barcodes including *YunSar* viruses are presented in Figure 4. Among them, there are five GB barcodes SC, four GB barcodes S5, and one GB barcode S3. The results revealed three different evolutionary histories for 5′, central, and 3′ genomic regions. The 5′ and central regions indicated a close relationship between *YunSar* and *SCoV2rC s.l.* (14% ≤ MBP ≤ 23% and 16% ≤ MBP ≤ 31%, respectively). However, a small zone of the 5′ region, located between pos. 1150 and 3300 showed a sister-group relationship between *YunSar* and Rc-o319 (10% ≤ MBP ≤ 12%), but the high nucleotide distance (33.0%) rather suggests a long-branch attraction artefact. The region located between pos. 10,900 and 12,700 appeared more similar to MjGuangxi (12% ≤ MBP ≤ 14%), with a nucleotide distance of 17.4%. By comparison, the nucleotide distances calculated for the GRPSs of the central region were much smaller, suggesting a more recent separation: between 7.3 and 8.1% with *SCoV2rC* (pos. 15,600–16,500) and between 9.7 and 11.0% with MjGuangdong (pos. 13,550–15,250 and 18,000–19,700). In the 3′ region (pos. 21,500–28,800), the four *YunSar* genomes appeared highly divergent from all other sarbecoviruses (29.7–30.1%), whereas other genomes found in Asia (11% ≤ MBP ≤ 21%) were found more similar to each other (*SCoVrC versus SCoV2rC s.l.* + Rc-o319 = 17.8–22.0%). All these comparisons, therefore, indicate that the common ancestor of *YunSar* has emerged from recombination between divergent parental viruses.

### 3.5. GB Barcodes of Bipartitions Including RecSar Viruses

The three *RecSar* viruses showing evidence of past recombination between *SCoVrC* and *SCoV2rC* are RpPrC31, RsZXC21 and RsZC45 [26,27]. Although *RecSar* appeared as a monophyletic group in the tree of Figure 3, the GB barcode associated with this node (bipartition N° 39; 14% ≤ MBP ≤ 22%) belongs to the S3 category, meaning that the phylogenetic support was only found in the 3′ region of the genome alignment (2 GRPSs in pos. 19,750–21,800 and 22,950–26,500).

All the 30 GB barcodes including *RecSar* viruses and supported by MBP_2000_ > 7% are shown in Figure 5 and Figure 6. The 7% threshold was chosen based on two opposing constraints: on the one hand, our desire for completeness results, and on the other, the lack of space in the figures (the 19 GB barcodes with MBP_2000_ < 7% were not shown). The results showed strong evidence for past recombination between *SCoVrC* and *SCoV2rC* lineages, as the analysis of nested GB barcodes revealed three different evolutionary histories for 5′, central, and 3′ genomic regions. In addition to the monophyly of *RecSar* viruses, the 3′ region provided support for their close relationship to RpYN06 within *SCoV2rC* (8% ≤ MBP ≤ 13%; 2 GRPSs in pos. 20,650–23,750 and 24,300–24,400). The 5′ region also showed that *RecSar* viruses belong to the *SCoV2rC* group (8% ≤ MBP ≤ 18%; 4 GRPSs in pos. 3050–4750, 5600–5750, 6600–7650, and 24,300–24,400): RpPrC31 appeared more closely related to RmYN02 (12% ≤ MBP ≤ 19%; 1 GRPS in pos. 2150–6600), whereas the position of RsZXC21 and RsZC45 was found to be unstable within the *SCoV2rC* group (see GB barcodes S5 in Figure 5). By contrast, the central region showed that the three *RecSar* viruses belong to the *SCoVrC* group (7% ≤ MBP ≤ 25%): RpPrC31 was found in the group of Southwest China (*SCoVrC*-SW) (10% ≤ MBP ≤ 20%; 2 GRPSs in pos. 12,900–14,150 and 15,600–18,450), where it appeared in the *SCoVrC*-SW2 subgroup (6% ≤ MBP ≤ 14%; 2 GRPSs in pos. 12,900–14,800 and 16,850–18,400); RsZXC21 and RsZC45 were found in the group of Central and East China (*SCoVrC*-CE) (4% ≤ MBP ≤ 20%; 1 GRPS in pos. 14,200–19,500), in which they appeared into a clade containing three viruses from Hong-Kong (RsHKU3-1, -7, and -12) and RmoLongquan140 from Zhejiang (15% ≤ MBP ≤ 28%; 1 GRPS in pos. 12,200–19,700), the latter virus being their sister-group (8% ≤ MBP ≤ 13%; 1 GRPS in pos. 15,200–18,300). All these results, therefore, indicate that the ancestor of RpPrC31 and the common ancestor of RsZXC21 and RsZC45 have emerged from two independent events of recombination between *SCoVrC* and *SCoV2rC* parental viruses.

### 3.6. GB Barcodes of Bipartitions Containing SARS-CoV-2

Most GB barcodes including SARS-CoV-2 and supported by MBP_2000_ > 7% are presented in Figure 7 (25/33 GB barcodes; the eight barcodes N° 17, 37, 41, 46, 54, 117, 120, and 129 were shown in previous figures).

The results revealed the mosaic origin of SARS-CoV-2, as several genomic parts of SARS-CoV-2 were found to share recent ancestry with several viruses recently described from *Rhinolophus* bats collected in Yunnan. The close relationship between SARS-CoV-2 and RmYN02 + RpYN06 (9% ≤ MBP ≤ 21%) was supported by three GRPSs (in pos. 1–1650, 7950–8700, and 17,950–20,200) showing between 1.5 and 3.0% of nucleotide divergence. Several other genomic regions provided support for a sister-group relationship of SARS-CoV-2 with either RaTG13 (14% ≤ MBP ≤ 21%; 2 GRPSs in pos. 4650–4950 (2.3%) and 20,300–23,300 (7.2%)), RmYN02 (8% ≤ MBP ≤ 11%; 1 GRPS in pos. 19,500–20,150 (1.8%)), or RpYN06 (7% ≤ MBP ≤ 8%; 1 GRPS in pos. 9150–10,350 (1.4%)). The SARS-CoV-2 was also found to have a common ancestry with the following groups of viruses from Yunnan: RmYN02 + RpYN06 + RaTG13 (7% ≤ MBP ≤ 17%; 2 GRPSs in pos. 18,500–20,250 (2.2–2.9%) and 24,800–26,950 (2.6–3.2%)), and RmYN02 + RpYN06 + RpPrC31 (6% ≤ MBP ≤ 15%; 2 GRPSs in pos. 1–2900 (2.7–3.6%) and 6250–7700 (1.6–2.3%)).

Several other genomic regions of SARS-CoV-2 were found to have a more ancient ancestry with several viruses from Yunnan, such as RmYN02 + RpYN06 + RaTG13 (7% ≤ MBP ≤ 17%; 2 GRPSs in pos. 18,500–20,250 (2.2–2.9%) and 24,800–26,950 (2.6–3.2%)), and RmYN02 + RpYN06 + RpPrC31 (6% ≤ MBP ≤ 15%; 2 GRPSs in pos. 1–2900 (2.7–3.6%) and 6250–7700 (1.6–2.3%)). Three GRPSs showed a common ancestry between SARS-CoV-2 and four viruses (7% ≤ MBP ≤ 18%), three from Yunnan (RmYN02 + RpYN06 + RaTG13) and one from Cambodia (RShSTT200): in pos. 7950–8350 (1.0–3.5%), 14,100–16,350 (1.9–2.5%), and 24,750–25,700 (3.4–5.0%). Among bipartitions with MBP_2000_ < 7% (not shown in Figure 7), we also found some traces of common ancestry between SARS-CoV-2 and viruses from Southeast Asia in pos. 11,200–12,450 (RShSTT200 + RacCS203; MBP_2000_ = 5%; distance = 2.2–2.4%) and 24,150–24,200 (RShSTT200; MBP_2000_ = 5%; distance = 7.8%), but these regions remains more divergent and represent much less phylogenetic signal than the sum of regions related to viruses from Yunnan (4.5 vs. 52.8% of the total genomic alignment). We did not find any GRPS shared only between SARS-CoV-2 and pangolin viruses, MjGuangdong and/or MjGuangxi.

### 3.7. Length Estimation of GRPSs Involving Viruses of the SCoV2rC s.l. Lineage

The succession of recombination events between *Sarbecovirus* genomes over time can result in a high fragmentation of phylogenetic signals distributed along the genomic alignment. In theory, the phylogenetic signal informing an ancient recombination (black B1 fragment in the recombinant genome R1 of Figure 8) can be partially lost when subsequent recombination has occurred in a nested location (blue B2 fragment in the recombinant genome R1′ of Figure 8). Such nested recombination events can greatly complicate the identification of ancient recombinant fragments. Therefore, we selected only the GB_2000_ barcodes showing a single GRPS to estimate the median size of putative recombinant fragments in the *SCoV2rC s.l*. lineage. For that purpose, we analyzed all the 183 bipartitions of the SWB_2000_ analysis including at least one of the 11 viruses of the *SCoV2rC s.l.* lineage and the 80 bipartitions showing a single GRPS (e.g., bipartitions N° 26, 100, and 127 in Figure 7) were selected to estimate GRPS lengths. The results revealed a high variation of GRPS lengths, from 50 nt to 8300 nt, and the median length was calculated to be 875 nt.

## 4. Discussion

### 4.1. Further Evidence for Four Divergent Sarbecovirus Lineages in Asia

Several studies have shown evidence for multiple recombination events during the evolutionary history of sarbecoviruses [3,28]. Consequently, different regions of recombinant genomes can bring discordant phylogenetic signals (Figure 8). For that reason, several phylogenetic trees based on different genes are generally compared in genomic studies on sarbecoviruses [8,9,16,17,29]. However, the reconstruction of separate gene trees implicitly assumes that recombination occurs between genes. Our results confirmed that this assumption is wrong as the high fragmentation of the phylogenetic support indicates that recombination has occurred everywhere in the genome and independently of the start and stop codons of ORFs (see also Appendix A). In a recent study, Boni et al. [3] have developed a two steps approach for phylogeny: firstly, recombination breakpoints were detected in the genomes using 3SEQ and GARD methods [30,31], and secondly, the genomic regions more likely to be non-recombinant were selected for tree reconstruction and molecular dating. Although these methods can be used to determine the regions less impacted by recent recombination, they cannot guarantee an efficient discovery of ancient recombination breakpoints.

In this study, we adopted a third phylogenetic approach in which the SuperTRI method [19] was used to construct an SB consensus tree showing the most reliable (repeated) relationships found in the SWB analysis. Our underlying assumptions for this approach can be formulated as follows: closely related viruses share robust phylogenetic signals in several large genomic regions, whereas more distantly related viruses have fewer genomic regions in common, most of them being reduced in size due to successive recombination events in overlapping or nested locations (Figure 8). It is important to note that the four SB trees constructed from the SWB analyses based on four different window sizes were found to be very similar, and that the topology remained very stable when recombinant viruses between divergent lineages (*RecSar* and *YunSar*) were removed from the analyses (Figure 3 vs. Appendix A). Such stability of phylogenetic relationships deeply contrasts with the discordant trees recently published for the subgenus *Sarbecovirus* [16,18].

Our synthesis tree provided high support for the four following *Sarbecovirus* lineages in Asia (Figure 3): (i) *SCoVrC*, including SARS-CoV and many bat viruses collected in different provinces of China; (ii) *SCoV2rC s.l.*, the lineage uniting *SCoV2rC* (SARS-CoV-2 + bat viruses from Yunnan and Southeast Asia), the two pangolin viruses (MjGuangdong and MjGuangxi from unknown origins in Southeast Asia), and the three *RecSar* viruses discovered in bats from two Chinese provinces, Yunnan (RpPrC31) and Zhejiang (RsZXC21 and RsZC45); (iii) the four *YunSar* viruses from Yunnan; and (iv) the Japanese virus Rc-o319. The *SCoVrC* and *YunSar* lineages were found to be monophyletic in recent studies [16,18]. By contrast, *SCoV2rC s.l.* was found to be paraphyletic in most trees shown in Zhou et al. [16] due to the inclusive placement of *YunSar*, which appeared as the sister-group of either MjGuangxi or MjGuangdong. In Guo et al. [18], both *SCoV2rC s.l.* and *YunSar* were found monophyletic, but *YunSar* appeared highly divergent from other sarbecoviruses. Our analysis of GB barcodes revealed that these topological incongruences resulted from past recombination events between *YunSar* and *SCoV2rC s.l.* lineages (involving MjGuangdong, MjGuangxi, and *SCoV2rC*; see GB barcodes N° 71, 90, and 145 in Figure 4). Importantly, all the four *Sarbecovirus* lineages supported by our phylogenetic analyses were found to have different synonymous nucleotide composition (SNC) in third codon-positions, dinucleotides, and degenerate codons [17]. Currently available data suggest that they have evolved in different biogeographic regions and/or distinct *Rhinolophus* reservoirs: Rc-o319 in Japan (reservoir: *R. cornutus*), *SCoVrC* in China (main reservoir: *R. sinicus*), *SCoV2rC s.l.* in Southeast Asia and Yunnan (main reservoir: several *Rhinolophus* species that do not hibernate, such as *R. acuminatus, R. malayanus*, *R. shameli*), and *YunSar* in Yunnan and maybe in Southeast Asia (most likely reservoir: *R. stheno*; see below for more explanations) [6,17].

### 4.2. RdRp Selection of Recombinant Genomes in Bat Reservoirs

Since the three divergent lineages *SCoVrC*, *SCoV2rC*
*s.l.,* and *YunSar* can be found at least occasionally in sympatry in the caves of Yunnan, recombination between them may have occurred in the Chinese province at multiple times during the evolutionary history of sarbecoviruses. Our analyses of nested GB barcodes showed strong evidence that the three *RecSar* viruses (RpPrC31, RsZC45, and RsZXC21) have emerged through two independent events of recombination between *SCoVrC* (donor parental genome; Figure 8) and *SCoV2rC* (acceptor parental genome, the one accepting a genomic fragment from the donor parental genome), confirming some results from previous studies [3,26,27]: one resulted in the ancestor of RpPrC31, and the other led to the ancestor of RsZC45 and RsZXC21.

The RpPrC31 virus was recently described from an intestinal sample of *R. pusillus* (subspecies *blythi*) collected in 2018 in Yunnan [27]. The analysis of GB barcodes showed that its genome contains highly discordant phylogenetic signals (Figure 6): based on both 5′ and 3′ genomic regions, it appeared related to bat *SCoV2rC* viruses from Yunnan (RmYN02 and RpYN06); based on the central genomic region, it appeared within *SCoVrC*-SW2, the clade including SARS-CoV and several bat viruses detected in Yunnan and three other provinces of Southwest China (Guangxi, Guizhou, and Sichuan). Biogeographically, the most parsimonious scenario therefore implies that recombination between *SCoVrC* (donor) and *SCoV2rC* (acceptor) genomes took place in Yunnan or adjacent regions. Such a scenario is also corroborated by the fact that the ecological niches of *SCoVrC* and *SCoV2rC* overlap only in the zone including southern Yunnan, northern Laos, and northern Vietnam [6].

The two viruses RsZXC21 and RsZC45 were discovered in *R. sinicus* samples collected in the Zhejiang province in 2015 and 2017, respectively [26]. As for the RpPrC31 genome, the analysis of GB barcodes showed that the genomes of RsZC45 and RsZXC21 contain highly discordant phylogenetic signals: based on both 5′ and 3′ regions, their common ancestor appeared related to bat viruses from Yunnan (RaTG13, RmYN02, RpYN06, and RpPrC31); based on the central genomic region, it was found within *SCoVrC*-CE as the sister-group of RmoLongquan140 from Zhejiang (Figure 5). Biogeographically, it can be therefore proposed that the parental *SCoV2rC* strain (acceptor genome) of the common ancestor of RsZXC21 and RsZC45 may have originated in Yunnan a few generations ago, and that recombination with a virus related to RmoLongquan140 (donor genome) occurred in the Zhejiang province.

The *YunSar* viruses were recently described from one *R.* affinis and seven *R. stheno* sampled in Mojiang County in 2015 [18] and from one *R. stheno* and two *R. malayanus* sampled in Mengla county (southern Yunnan) between 2019 and 2020 [16]. However, the re-analyses of SRA data have shown that all the three viruses published by Zhou et al. [16] were collected from *R. stheno* (AH, paper in preparation), meaning that RmYN05 and RmYN08 should be renamed RstYN05 and RstYN08, respectively. The *YunSar* genomes were found to evolve with higher rates of substitutions and with atypical asymmetric mutational constraints [17], suggesting a bat reservoir different from that of the two other lineages currently found in Yunnan, *SCoVrC* and *SCoV2rC*. Obviously, the species *R. stheno* seems to be the most likely reservoir candidate for this lineage, as *YunSar* was found with higher prevalence in this species (10 out of 11 samples). Another interesting point is that currently available data on the distribution of *R. stheno* suggest that the population from Yunnan may be isolated from other populations found in southern Vietnam, Malaysia, and Indonesia [13]. The analysis of GB barcodes showed that *YunSar* genomes contain highly discordant phylogenetic signals: based on the 5′ region, they appeared as the sister-group of *SCoV2rC s.l.*; based on the 3′ region, they were found divergent from all other *Sarbecovirus* lineages; and based on the central region, they appeared closely related to either MjGuangxi, MjGuangdong or the common ancestor of *SCoV2rC* (Figure 4). The results therefore suggest that the central genomic region has been acquired through recombination with several *SCoV2rC s.l.* viruses.

It is relevant to note that the central region from a donor parental genome has been selected in all current sarbecoviruses showing evidence of past recombination between divergent lineages: the recombinant pattern *SCoV2rC*-*SCoVrC*-*SCoV2rC* has been selected in the Zhejiang ancestor of RsZC45 and RsZXC21 and independently in the Yunnan ancestor of RpPrC31; and the recombinant pattern *YunSar*-*SCoV2rC s.l.*-*YunSar* has been selected in the Yunnan (or possibly Southeast Asian) ancestor of *YunSar*. The convergent acquisition of three similar patterns of recombination is indicative of strong selective pressure in the central genomic region. In agreement with that hypothesis, Figure 2A shows that the number of IS is lower in the central region than in the 5′ and 3′ regions, indicating that the genes of the central region have evolved under stronger selective pressure. Importantly, the central region contains the protein-coding sequence of the *RdRp*, which plays a crucial role in the replication and transcription of the viral RNA genome [32], and has evolved under strong negative selection throughout the COVID-19 pandemic [25].

Both viruses RsZC45 and RsZXC21 were found in *R. sinicus* bats from Zhejiang, suggesting their common ancestor originated in the same species and province. As previously exposed, we hypothesized that recombination occurred in Zhejiang between an acceptor *SCoV2rC* genome, coming originally from Yunnan, and a donor *SCoVrC*-CE genome related to RmoLongquan140. Importantly, *R. sinicus* is likely to be the main reservoir species for *SCoVrC*, as most viruses of this lineage have been sampled in this species [7]. Moreover, the geographic range of *R. sinicus* [13] includes Zhejiang and most other provinces of China, which fits very well with the ecological niche recently inferred for *SCoVrC* [6]. Taken together, these elements imply that the ancestor of RsZC45 and RsZXC21 has emerged in a *SCoVrC* environment, both in terms of geography and bat reservoir, and that the selection of an *RdRp* gene of the *SCoVrC* type (previously adapted to *R. sinicus*) was determinant for its viral replication and proliferation in the bat species in which recombination occurred, i.e., *R. sinicus*. Based on the observed recombination patterns, we similarly propose that the ancestor of RpPrC31 has also been selected in a *SCoVrC* environment, while the ancestor of *YunSar* has originated in a *SCoV2rC s.l.* environment. The hypothesis involving a strong host-dependent selection of the viral RNA-dependent RNA polymerase is also corroborated by the phylogeographic patterns found in both *SCoVrC* and *SCoV2rC* lineages (Figure 3 and Appendix A): within *SCoVrC*, we found a basal dichotomy separating two geographic groups, *SCoVrC*-SW in Southwest China and *SCoVrC*-CE in Central and East China; within *SCoV2rC*, all bat viruses sampled in Yunnan were grouped (*SCoV2rC*-YU), with the two bat viruses from southern Southeast Asia (Cambodia and Thailand) at the outside.

### 4.3. SARS-CoV-2 Is a Mosaic Genome Closely Related to Bat Viruses from Yunnan

There are several hypotheses regarding the origin of SARS-CoV-2, including direct transmission from horseshoe bats to humans, indirect transmission via the Sunda pangolin or another intermediate host species [5], or a laboratory escape [33]. The two viruses previously described from Sunda pangolins [31,34], here named MjGuangdong and MjGuangxi, were found to share a similar SNC (characterized by the highest percentage of A nucleotide at third codon-positions), which was different from that observed in SARS-CoV-2 and bat *SCoV2rC*s (characterized by the highest percentages of U nucleotide and lowest percentages of C nucleotide at third codon-positions) [17]. Since MjGuangdong and MjGuangxi were never found as sister-groups, Hassanin [17] concluded that the two independent host switches from *Rhinolophus* bats to pangolins have led to convergent mutational constraints and that SARS-CoV-2 has emerged directly from a bat virus, without a pangolin intermediate host. In agreement with this view, our analysis of GB barcodes (Figure 7) showed that many genomic regions of SARS-CoV-2 were found closely related to viruses recently described from *Rhinolophus* bats collected in Yunnan, including RmYN02, RpYN06, and RaTG13. Our results therefore suggest that the ancestor of SARS-CoV-2 emerged in bats from Yunnan, a hypothesis also corroborated by the consensus tree of our SuperTRI analyses (Figure 3), in which SARS-CoV-2 appeared as the sister-group of RmYN02 + RpYN06 from Yunnan, with RaTG13 from Yunnan in more basal position. The same phylogeographic pattern was recovered with more robustness (higher MBP values) when recombinant viruses between divergent lineages, *RecSar* and *YunSar*, were removed from the analyses (Appendix A). Although an origin in Yunnan is currently the best hypothesis, northern regions of Southeast Asia adjacent to Yunnan, such as northern Myanmar, northern Laos, and northern Vietnam, cannot be completely ruled out. In agreement with that hypothesis, viruses closely related to SARS-CoV-2 were recently discovered in northern Laos [35]. However, to date, none of these viruses were found to contain the insertion of four amino acids (PRRA motif) at the cleavage site of the spike protein, which is known to have a critical role in SARS-CoV-2 infection and pathogenesis [36].

### 4.4. Are CircRNAs Involved in Homologous Recombination?

Circular RNAs (circRNAs) form a large class of RNA molecules widespread in eukaryotes [37]. They play important functions in gene regulation as they can have interactions with RNA-binding proteins, serve as translation templates, and act as microRNA sponges or transcriptional regulatory factors [37,38,39]. DNA-genome viruses are also capable of generating circRNAs [40,41], and recent studies have also reported that circRNAs can be generated from RNA-genome viruses [42,43]. In SRA data generated from three pangolin lung samples, Hassanin et al. [42] found 10 circRNAs of 278-776 nt derived from the MjGuangdong virus. In addition, thousands of circRNAs encoded by SARS-CoV and SARS-CoV-2 were recently described in Cai et al. [43]. They were detected in all genomic regions, and most of them had lengths ranging from 200 nt to 2000 nt, some of them being much longer. Although the roles of viral circRNAs remain poorly known, Ungerleider et al. [41] have found that the Epstein Barr virus expressed circRNAs during latent infection and under reactivation conditions.

By assuming that GB barcodes showing a single isolated GRPS represent a good proxy of phylogenetic signals inherited by recombination, we calculated a median length of 875 nt for the 80 bipartitions involving at least one virus of the *SCoV2rC s.l*. lineage. The value is very similar to the median length of viral circRNAs recently estimated by Cai et al. [43] for SARS-CoV and SARS-CoV-2, with 812 nt and 791 nt, respectively. These comparisons therefore suggest that viral circRNAs may be involved in the mechanism of genomic recombination. Three other arguments are in favor of this hypothesis. Firstly, most representations of the copy-choice model assume that the product of RNA recombination is composed of two genomic regions: the 5′ region is inherited from one parental genome, and the 3′ region is inherited from another parental genome [1,44,45]. However, such a recombination pattern is generally not observed in GB barcodes (except for some rare bipartitions, such as nested bipartitions N° 41 and N° 54 in Figure 5, which are however compatible with the ’circRNA model’ of Figure 8), suggesting that a different mechanism operates in the case of sarbecoviruses. Indeed, most GB barcodes showed a pattern with one or several GRPS(s) (e.g., bipartitions N° 9, 17, 18, 32, 86, 117, 154 in Figure 5), indicating that the recombination mechanism rather involves a full-length acceptor parental genome and the fragment(s) (potentially circular) from a donor parental genome (as assumed by the ’circRNA model’ of Figure 8). Secondly, viral circRNAs are expressed in the late stage of viral infection [41,43], and their circular nature confers much more stability than linear RNAs [46], suggesting that they can stay in latency in the bat host cells for a longer period. This makes the co-occurrence of two different viral sequences more likely in the same bat cell (i.e., circRNAs derived from an ancient infection and full-length RNA genomes resulting from a new infection), a *sine qua non* condition for genomic recombination. Thirdly, many studies have concluded that RNA secondary structures are essential to promote template switching of the RdRp during replication [1,47,48]. Importantly, all circRNAs generated from the MjGuangdong pangolin virus were found to form highly stable secondary structures [42]. These elements, therefore, suggest that template switching may be caused by RdRp interactions with the secondary structure of viral circRNAs.

## Figures and Tables

**Figure 1 viruses-14-00440-f001:**
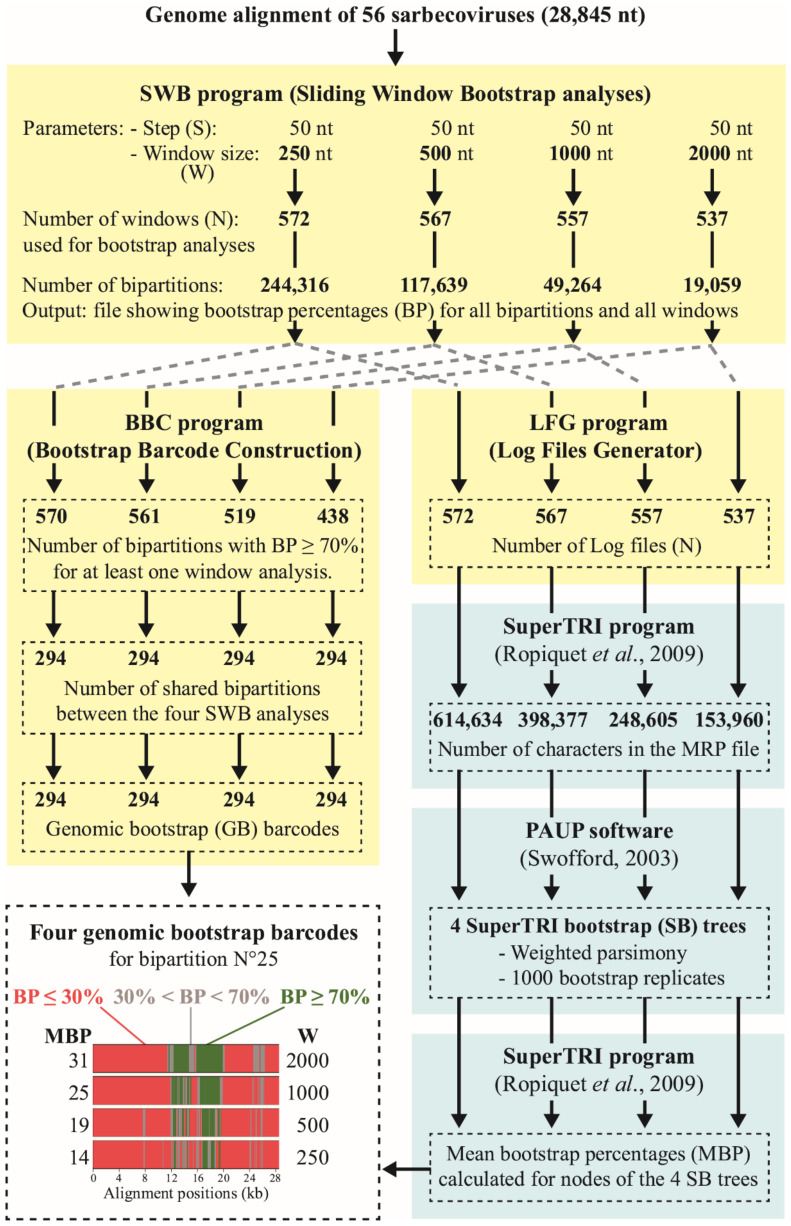
Diagram describing the programs used to construct genomic bootstrap barcodes and phylogenetic trees. The programs SWB, BBC, and LFG were specially written for this study (highlighted in pale yellow). The SWB program was designed to generate bootstrap bipartitions from a sliding window of a specific width (W) moving in steps of 50 nt along the genome alignment. With W sets to 250 nt, N = 572 RAxML bootstrap analyses [23] were conducted (using a GTR+G model and 100 replicates), and the bootstrap percentages (BP) calculated in each of the 572 analyses were reported in a CSV file for all bipartitions (nodes). Then, each of the four SWB files generated with four different window sizes (W = 250, 500, 1000, or 2000 nt) was used as input in two additional programs, BBC and LFG. The BBC program was used to construct GB barcodes for the 294 bipartitions shared between the four SWB analyses using the following color code: red for BP ≤ 30%; grey for 30% < BP < 70%; and green for BP ≥ 70%. The LFG program was used to produce the bootstrap log files, which were further transformed into an MRP (Matrix Representation with Parsimony) file with the SuperTRI v57 program [19]. In the MRP file, each character represents a bipartition with its assigned BP calculated in one of the window bootstrap analyses (e.g., N = 572 with W = 250 nt). The MRP file was then executed in PAUP version 4.0a [24] using bootstrap percentages as weights to construct the SuperTRI bootstrap 50% majority-rule consensus (SB) tree (weighted parsimony method, 1000 replicates). Finally, mean bootstrap percentages (MBP) were calculated under SuperTRI v57 for all nodes of the SB tree.

**Figure 2 viruses-14-00440-f002:**
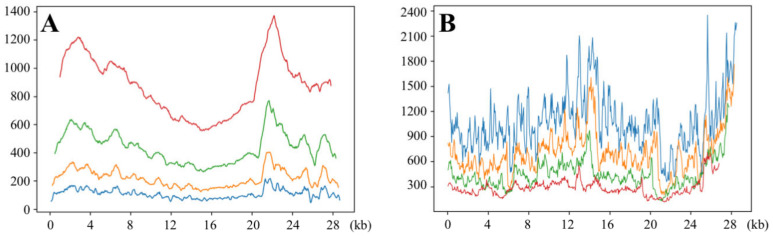
Number of informative sites (**A**) and bipartitions (**B**) found in the SWB analyses based on four window sizes. The alignment of 56 *Sarbecovirus* genomes was analyzed using four sliding window sizes: 250 nt (blue), 500 nt (orange), 1000 nt (green), and 2000 nt (red). The scale in abscissa shows the positions in the alignment.

**Figure 3 viruses-14-00440-f003:**
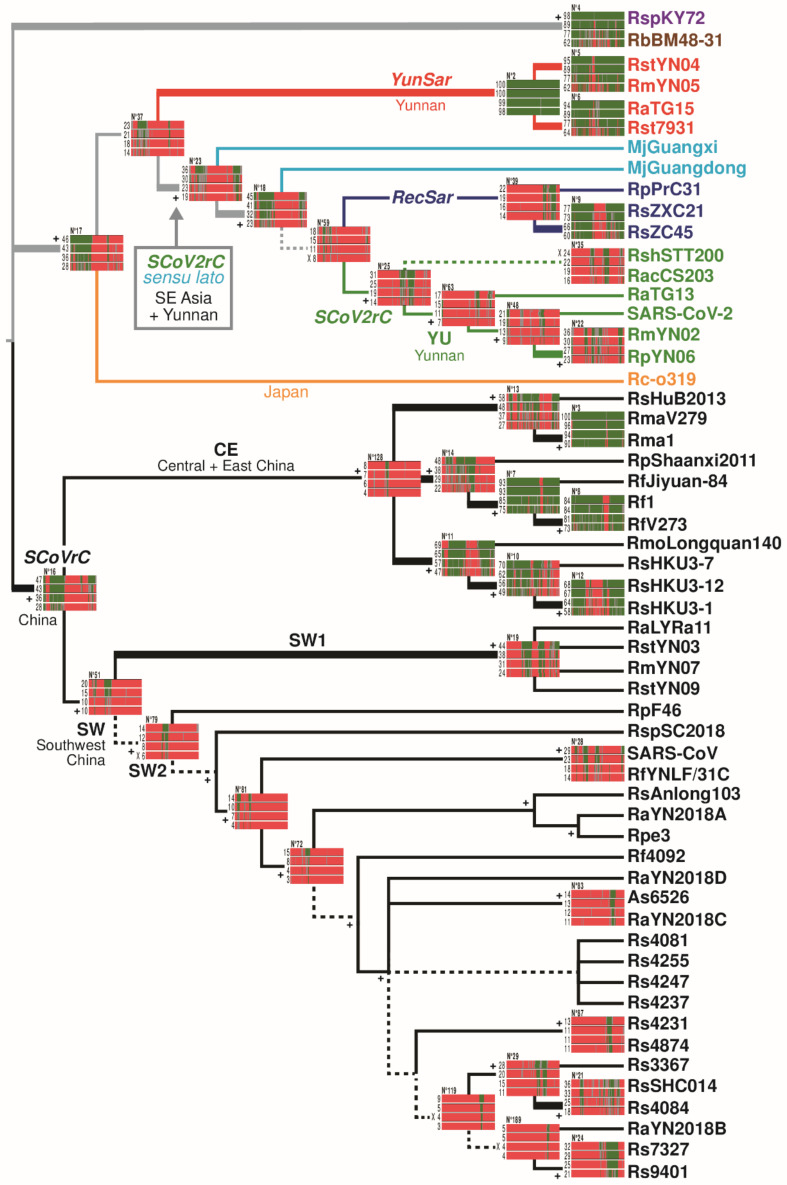
Synthesis tree showing the most reliable relationships reconstructed from SWB analyses of an alignment of 56 *Sarbecovirus* genomes. Bootstrap analyses were conducted on an alignment of 56 *Sarbecovirus* genomes (28,845 nt) using the SWB program and four different window sizes (250, 500, 1000, or 2000 nt). Then, the SWB outfiles were tranformed into bootstrap log files using the LFG program. The 572 bootstrap log files generated with a window size of 250 nt were coded by the SuperTRI v57 program [19] into an MRP file, which was then executed in the PAUP version 4.0a [24] to construct a SuperTRI bootstrap 50% majority-rule consensus (SB_250_) tree using the weighted parsimony method and 1000 bootstrap replicates. Three other SB trees were constructed using the same approach: SB_500_ with the 567 bootstrap log files generated with a window size of 500 nt; SB_1000_ with the 557 bootstrap log files generated with a window size of 1000 nt; and SB_2000_ with the 537 bootstrap log files generated with a window size of 2000 nt. The tree shown here is a 75% majority-rule consensus tree of SB_250_, SB_500_, SB_1000_, and SB_2000_ trees (shown in Appendix A). The nodes were recovered in all the four SB trees, except those indicated by dash branches, which were found monophyletic in only three of the four SB trees. The GB barcodes were constructed for all nodes showing at least one Bootstrap Percentage (BP) ≥ 70% in the four SWB analyses using different window sizes. Thick branches highlight the nodes supported by Mean Bootstrap Percentage (MBP) ≥ 33% with a window size of 2000 nt. The symbol “+” indicates the nodes also supported by the SWB analyses of a reduced alignment of 49 genomes, i.e., after exclusion of recombinant genomes between divergent lineages (*RecSar* and *YunSar*) (Appendix A). The colors of sarbecoviruses indicate to which group of synonymous nucleotide composition they belong [17]: black for SARS-CoV related coronaviruses (*SCoVrC*); green for coronaviruses related to SARS-CoV-2 (*SCoV2rC*); light blue for the two pangolin viruses (*PangSar*); dark blue for the three bat *RecSar* viruses showing evidence of genomic recombination between *SCoV2rC* and *SCoVrC*; red for the four divergent bat viruses from Yunnan (*YunSar*); and orange for the bat virus from Japan.

**Figure 4 viruses-14-00440-f004:**
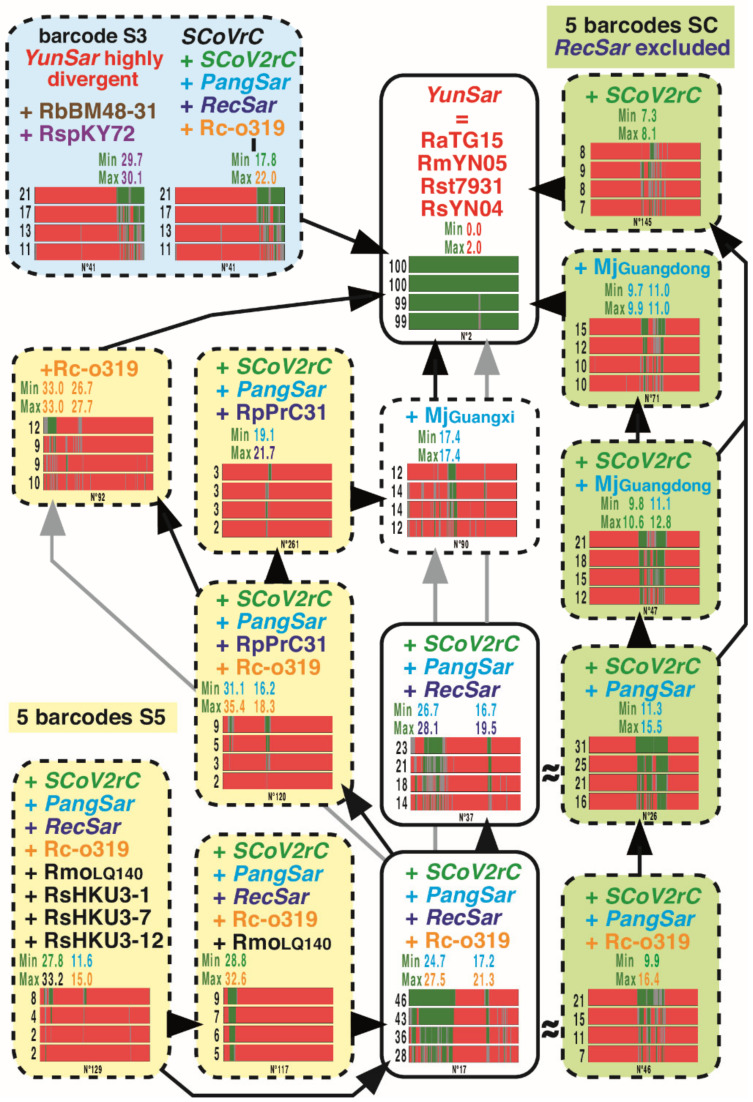
Genomic bootstrap (GB) barcodes generated for the bipartitions including *YunSar* viruses. For all the 15 bipartitions including the four *YunSar* viruses, four GB barcodes were constructed from SWB analyses based on the following window sizes: from bottom to top, 250, 500, 1000, and 2000 nt. In GB barcodes, genomic regions in green were supported by bootstrap percentages (BP) ≥ 70%, those in red by BP ≤ 30%, and those in grey by 30% < BP < 70%. For each GB barcode, the value at the left is the mean bootstrap percentage (MBP). The values above GB barcodes are minimum and maximum distances calculated in the genomic region(s) containing robust phylogenetic signal (GRPS) between *YunSar* and other viruses included in the bipartition. Four categories of GB barcodes were defined based on the genomic location of the phylogenetic support (BP ≥ 70%, green regions): pale yellow for the 5′ region; pale olive green for the central region; light blue for the 3′ region; and white when the phylogenetic support was distributed in different genomic regions. Only the bipartitions surrounded by a solid black line were found monophyletic in the tree of Figure 3. The bipartitions surrounded by a dashed black line represent hidden phylogenetic signals. As in Figure 3, the colors of sarbecoviruses indicate to which group of synonymous nucleotide composition they belong. The isosceles triangles show nested GB barcodes, the vertex point indicating the nested bipartition, i.e., the one containing fewer viruses. The symbol “≈” was used to show compatible bipartitions considering the recombinant nature of *RecSar* genomes.

**Figure 5 viruses-14-00440-f005:**
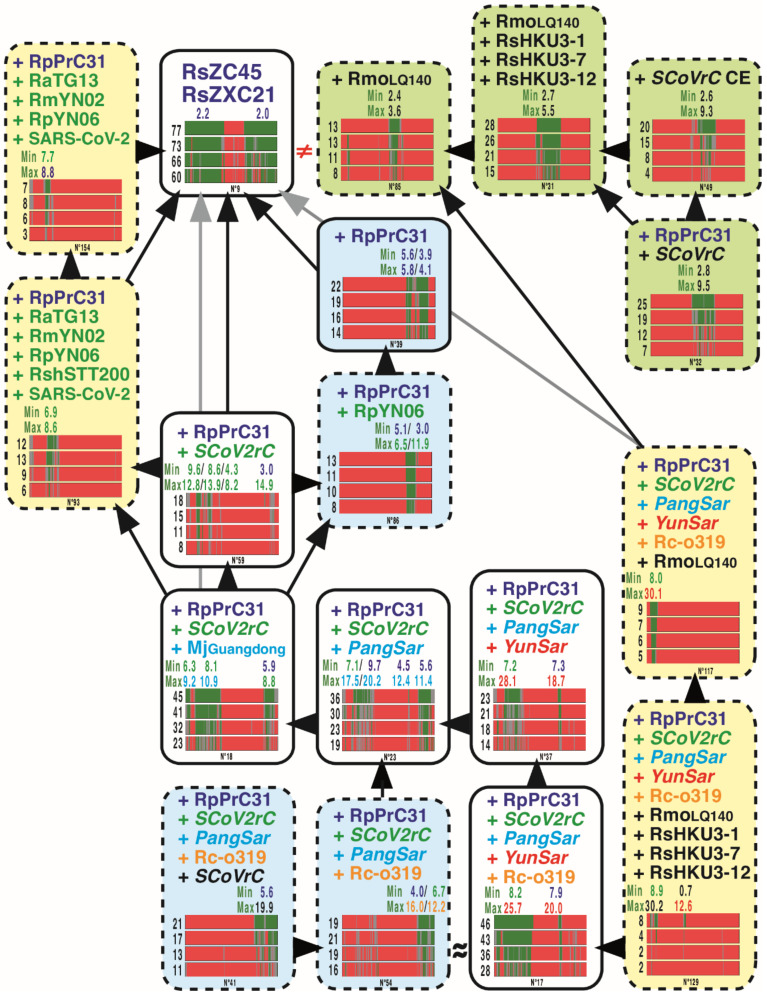
Genomic bootstrap (GB) barcodes generated for the bipartitions including RsZC45 and RsZXC21. The GB barcodes are shown for the 18 bipartitions including RsZC45 and RsZXC21 found in the SWB analyses based on four window sizes (250, 500, 1000, or 2000 nt) and supported by MBP_2000_ > 7%. For more details on GB barcodes, read the legend of Figure 4. Here, the GB barcodes with MBP_2000_ < 7% were not shown.

**Figure 6 viruses-14-00440-f006:**
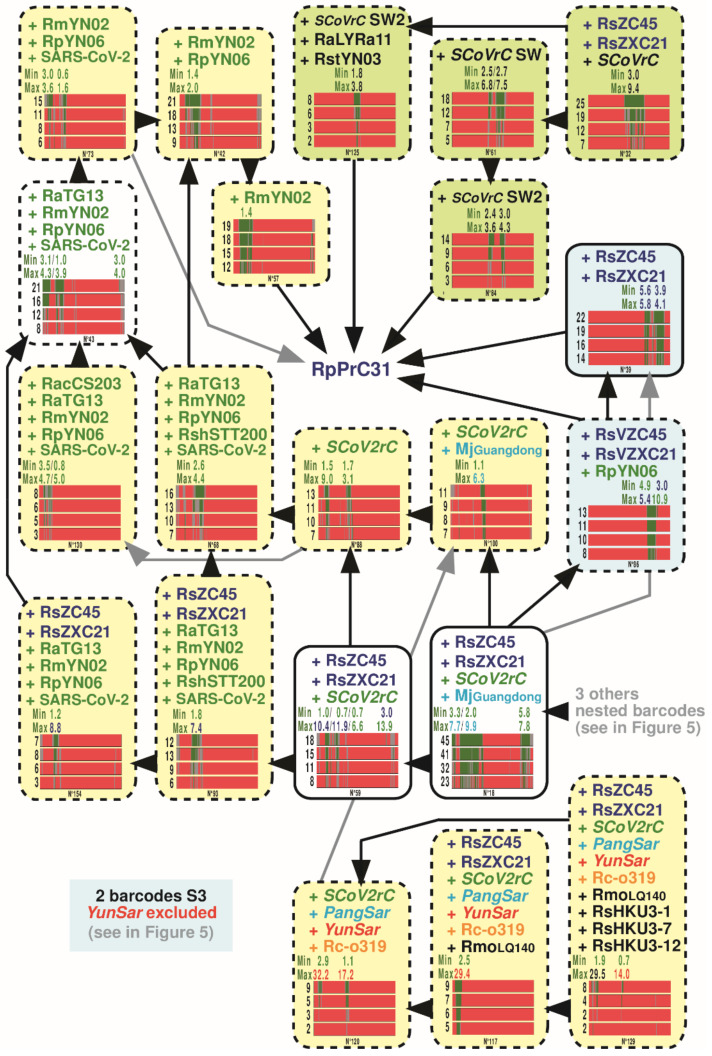
Genomic bootstrap (GB) barcodes generated for the bipartitions including RpPrC31. The GB barcodes were examined for the 26 bipartitions including RpPrC31 found in the SWB analyses based on four window sizes (250, 500, 1000, or 2000 nt) and supported by MBP_2000_ > 7%. For more details on GB barcodes, read the legend of Figure 4. Here, the GB barcodes with MBP_2000_ < 7% were not shown.

**Figure 7 viruses-14-00440-f007:**
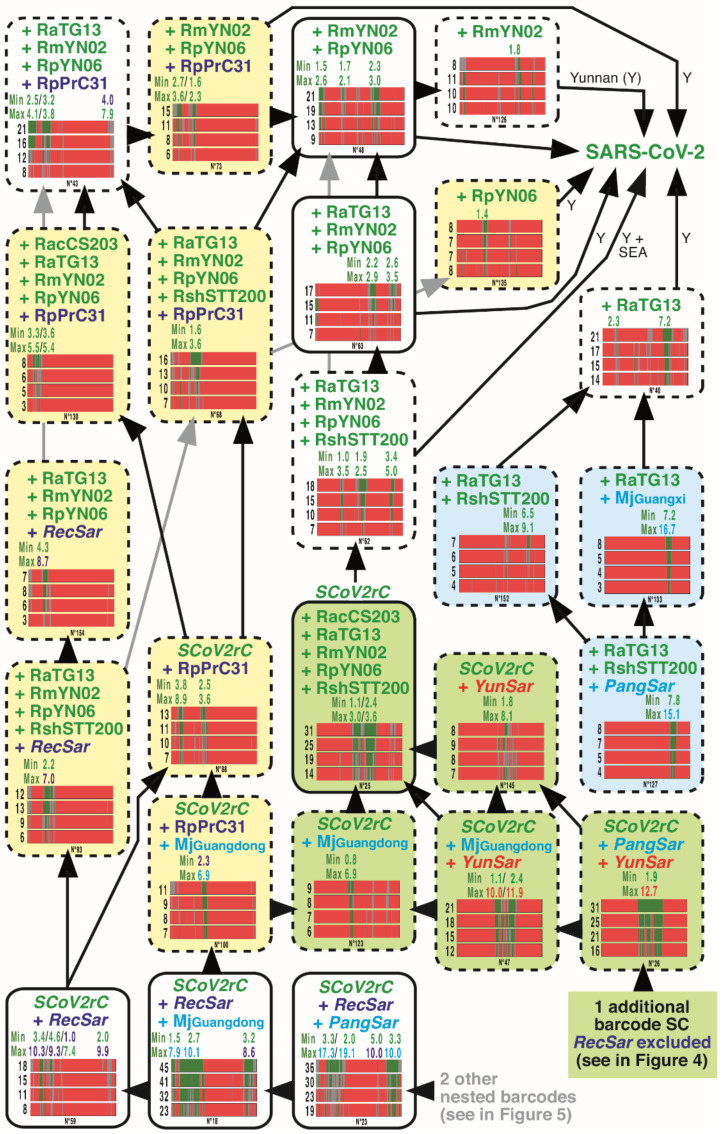
Genomic bootstrap (GB) barcodes generated for the bipartitions including SARS-CoV-2. The GB barcodes were examined for the 33 bipartitions including SARS-CoV-2 found in the SWB analyses based on four window sizes (250, 500, 1000, or 2000 nt) and supported by MBP_2000_ > 7%. The GB barcodes of the eight largest bipartitions (N° 17, 37, 41, 46, 54, 117, 120, and 129) are not shown; they are available in Figure 4, Figure 5 and Figure 6. For more details on GB barcodes, read the legend of Figure 4. Here, the GB barcodes with MBP_2000_ < 7% were not shown.

**Figure 8 viruses-14-00440-f008:**
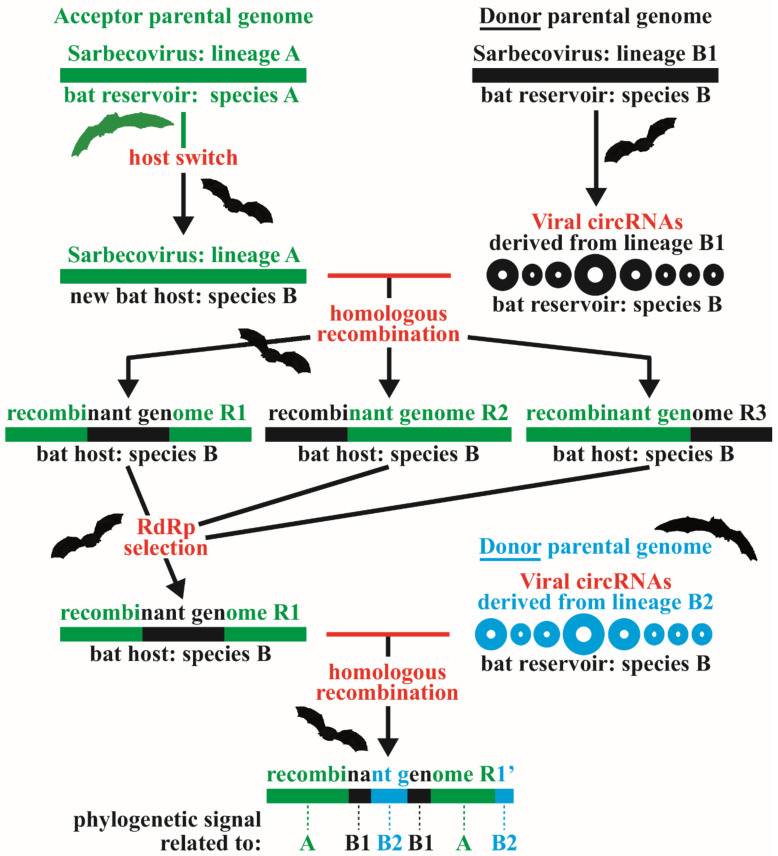
Model of recombination between different *Sarbecovirus* lineages. The two parental genomes are defined as acceptor and donor. We assumed that the donor parental genome was preserved in the host cell as circular RNA molecules (see Section 4.4), whereas the cell was subsequently infected with the acceptor parental genome. Then, the process of homologous recombination can generate different patterns of recombinant genomes, which are selected based on their capacity to replicate efficiently in the bat reservoir host, i.e., through the sequence of their RNA-dependent RNA polymerase (RdRp). According to this model, the accumulation of recombination events over time has led to mosaic genomes in which different regions can carry incongruent phylogenetic signals.

## Data Availability

The genome alignment, the four SWB output files, the four SuperTRI MRP files, the four SB trees (SB_250_, SB_500_, SB_1000_, and SB_2000_), and all the 1176 GB barcodes constructed in this study are available in the Open Science Framework (OSF) platform at https://osf.io/wjhyc/ accessed on 1 June 2021. The SWB, BBC, and LFG programs are available at https://github.com/OpaleRambaud/GBbarcodesproject accessed on 1 June 2021.

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
