# Peer review of "Genomic Bootstrap Barcodes and Their Application to Study the Evolution of Sarbecoviruses"

_viruses, 2022, doi:10.3390/v14020440_

Round 1

Reviewer 1 Report

Authors developed a sliding window bootstrap (SWB) method to generate genomic bootstrap barcodes to detect regions with phylogenetic relationships across recombinant genomes, and used to evaluate the method a dataset of 56 sarbecoviruses, including SARS-CoV and SARS-CoV-2. The method seems sound, but there are few issues that, if addressed, can make the manuscript stronger.

I am not sure I understand the concept of barcode. Barcodes are stretches of unique nucleotides used to make sequencing libraries. Can authors explain better what does barcode mean in this context?

Can the authors say what language the SWB, BBC, and LFG programs were written in? is the code available?

One major issue is that the authors did not explain how this new approach differs from SplitsTree, GARD or other recombination algorithms already available? I think a performance and comparison is needed in order to understand the performance of the new method. Moreover, I think before diving into analysis of a unknown datasets, the authors should make a in silico recombinant dataset and see whether the algorithm is able to find the breakpoints, or alternatively use a known/published recombinant dataset and test that against SplitsTree, GARD or other recombination algorithms.

Another issue is that authors did not compare their results with the previously published analysis of Boni et al that was published last year and also looks at recombinant regions of a 68-genome sarbecovirus. They use 3 different methods. Authors should repeat this analysis and use the same dataset.

Reviewer 2 Report

The inference of the phylogenetic relationship among viral genomes is complicated by recombination between divergent viral genomes that are coinfected.  Since the recombinant viruses comprise bits and pieces of different ancestries, the mosaic genomic regions influence the reconstruction of the phylogenetic relationship.  In this manuscript, the authors describe a method that identifies parts of the genomic regions that are less likely influenced by recombination and use them for inferring the phylogeny.  This was achieved by calculating the bootstrap confidence for various bits of the genomes (small and large), and those with high BS values were chosen for inferring the phylogeny.  I think this is a simple but interesting method worth publishing in Genes.  I have a few comments that might help to improve the quality of the manuscript.

  1. Although the number of informative sites (IS) useful for phylogenetic reconstruction has been mentioned throughout the manuscript, I would like to see the proportion of the viral genome alignment that constitutes/belongs to the IS. For instance, based on Figure 1, for the 2000kb window size, the length of IS (BP > 70%) is ~8kb (as it spans from 12kb-20 kb of the genomes).  Hence this constitutes 28% of the genome (8/29).  It is good to see the proportion of informative sites present in the genomes that were used to calculate the BP for each node.  Therefore, instead of mentioning the number of IS alone, providing the proportions of the genome will be a useful statistic for the readers to understand the method better.

  1. The genomic regions that are highly variable or mutable will also receive low bootstrap values due to weak phylogenetic signals. The authors need to discuss this issue and explain how their method distinguishes these regions from the recombinant regions.

  1. Figure 3 shows that the proportion of IS for some of the deep basal nodes is much higher than those of the shallow nodes. For instance, for the 2000kb window size, the proportion of green (BP > 70%) for the basal node N16 appears to be much higher than the shallow nodes N29, N189, N119, N72, etc.  A similar pattern can be seen comparing the basal node N17 and other shallow

Round 2

Reviewer 1 Report

authors responses and edits are satisfactory.